# API Content and Blend Uniformity Using Quantum Cascade Laser Spectroscopy Coupled with Multivariate Analysis

**DOI:** 10.3390/pharmaceutics13070985

**Published:** 2021-06-29

**Authors:** Vladimir Villanueva-López, Leonardo C. Pacheco-Londoño, Reynaldo Villarreal-González, John R. Castro-Suarez, Andrés Román-Ospino, William Ortiz-Rivera, Nataly J. Galán-Freyle, Samuel P. Hernandez-Rivera

**Affiliations:** 1ALERT DHS Center of Excellence for Explosives Research, Department of Chemistry, University of Puerto Rico, Mayagüez, PR 00681, USA; vladimir.villanueva@upr.edu (V.V.-L.); leonardo.pacheco@unisimonbolivar.edu.co (L.C.P.-L.); jhon.castro@upr.edu (J.R.C.-S.); william.ortiz5@upr.edu (W.O.-R.); 2Pharmaceutical Chemistry Department, School of Basic and Biomedical Sciences, Universidad Simón Bolívar, Barranquilla 080002, Colombia; 3AudacIA Center, Universidad Simón Bolívar, Barranquilla 080002, Colombia; reynaldovilla@gmail.com; 4Exact Basics Area, Universidad del Sinú, Unisinú, Cartagena 130015, Colombia; 5Engineering Research Center for Structured Organic Particulate Systems (C-SOPS), Department of Chemical and Biochemical Engineering, Rutgers, The State University of New Jersey, Piscataway, NJ 08854, USA; adr112@rutgers.edu

**Keywords:** quantum cascade laser, PAT, infrared spectroscopy, blend uniformity, content uniformity

## Abstract

The process analytical technology (PAT) initiative proposed by the US Food and Drug Administration (FDA) suggests innovative methods to better understand pharmaceutical processes. The development of analytical methods that quantify active pharmaceutical ingredients (APIs) in powders and tablets is fundamental to monitoring and controlling a drug product’s quality. Analytical methods based on vibrational spectroscopy do not require sample preparation and can be implemented during in-line manufacturing to maintain quality at each stage of operations. In this study, a mid-infrared (MIR) quantum cascade laser (QCL) spectroscopy-based protocol was performed to quantify ibuprofen in formulations of powder blends and tablets. Fourteen blends were prepared with varying concentrations from 0.0% to 21.0% (*w/w*) API. MIR laser spectra were collected in the spectral range of 990 to 1600 cm^−1^. Partial least squares (PLS) models were developed to correlate the intensities of vibrational signals with API concentrations in powder blends and tablets. PLS models were evaluated based on the following figures of merit: correlation coefficient (R^2^), root mean square error of calibration, root mean square error of prediction, root mean square error of cross-validation, and relative standard error of prediction. QCL assisted by multivariate analysis was demonstrated to be accurate and robust for analysis of the content and blend uniformity of pharmaceutical compounds.

## 1. Introduction

The high demand for medicine by patients with high-risk medical conditions is one of the pharmaceutical industry’s motivations to research alternatives to improve manufacturing processes. The development of analytical methods requiring minimal, or no sample preparation, a short analysis time, and high sensitivity is fundamental to minimizing specification products.

The traditional analytical methods for quantifying active pharmaceutical ingredients (API) include chromatographic [1,2,3,4] and optical techniques (UV–Vis) [5,6,7]. These methods have high reproducibility and low detection limits, although they involve destroying the samples or transporting them to the laboratory for further analysis. Alternatively, analytical methods based on infrared spectroscopy (IRS) and Raman spectroscopy in diffuse reflectance mode facilitate the analysis of several chemical compounds [8,9], including drug substances in different pharmaceutical products, from their manufacture until the final product validation, in a non-invasive way as well as in situ [10,11,12,13,14].

Near-infrared spectroscopy (NIRS) is becoming a mature approach for developing non-invasive methods in pharmaceutical applications, especially for API in powder blends and tablets in a diffuse reflectance configuration [15]. The method facilitates the inspection of multiple compressed tablets in a non-destructive fashion. However, one of the drawbacks of NIRS is its low sensitivity due to the low absorptivity of the overtones and combination bands present in this region of the electromagnetic spectrum; in addition to this, the signals are broad bands that require significant chemometric analysis to extract relevant information from the spectra. Furthermore, NIRS instruments have a weak radiation source, i.e., a globar, which produces a higher detection limit. Despite these limitations, NIRS is preferred for process analytical technology (PAT) applications due to its flexibility with acquisition modes such as transmittance and reflectance. [16].

The nature of mid-infrared (MIR) light restricts the analysis of condensate phases to a shallower depth of penetration than NIRS. In the MIR region, the molecules’ vibrational signatures are unique narrow bands, providing fingerprint identification; therefore, the specificity needed to identify the chemical compounds with high confidence is possible even in complex matrices. The high absorptivity of the vibrational bands limits the diffuse reflectance measurements, and in consequence, results in a low signal-to-noise ratio. To overcome these limitations, new technology with higher optical power could lead to more sensitive analytical methods.

Quantum cascade lasers (QCLs) are powerful semiconductor lasers that emit coherent high collimated MIR light [17] with a higher brightness than FTIR [18] and synchrotrons [19]. Some of the QCL applications reported include the long-distance detection of chemical compounds [20,21,22], sensing of proteins in aqueous solutions [23], quantification of explosives in soil [24] and petroleum in soil [25], and monitoring of chemical reactions [26]. These are a few of the challenging conditions wherein measurements were possible due to the high optical power of QCL. Due to the high brightness, QCL requires less integration time than FTIR and NIRS to operate in a higher signal-to-noise spectrum. Due to the high resolution of QCL, the analysis monitoring of gases [27,28] at extremely low concentrations with high selectivity is feasible, demonstrating the versatility of this instrument. Ostendorf et al. [29] have shown the capabilities of QCL in diffuse reflectance mode for the analysis of food quality, detection of the presence of molds in peanuts, and remote detection of explosives with back reflection measurements.

This study is the continuation of a proposed reproducible methodology using MIR QCL spectroscopy in a diffuse reflectance mode [30] with a high analytical sensitivity equivalent of 0.05% (*w/w*) API in the formulation, high repeatability (2.7% (*w/w*)), and high reproducibility (5.4% (*w/w*)). Three diodes were used to cover a spectral region of 990 to 1600 cm^−1^ and multivariate analysis (MVA) was used to quantify drug content and validate the API, ibuprofen (IBU). We compared the spectral differences between tablets and powders and showed the possibility of implementing them in both presentations. Moreover, we suggested that it is possible to use a single laser with a smaller spectral region which minimizes the system and scanning time. Several partial least squares (PLS) models were developed for each preprocessing treatment of the spectroscopic data. A discussion of the three best PLS models and the effect of the focal distance in the prediction error was presented. The proposed method could be implemented in real time, providing accurate data to enable control strategies.

## 2. Materials and Methods

### 2.1. Reagents and Materials

The materials used in this study included IBU as an API and excipients. API (≥98% GC grade) was purchased from Sigma-Aldrich (Millipore Sigma, St. Louis, MO, USA). Four excipients were used and blended with IBU to obtain powders and tablets in the concentration range tested. Lactose monohydrate (LM; granular; Meggle Pharma, MEGGLE Group, Wasserburg, Germany), microcrystalline cellulose (MCC; Vivapur^®^ 102, JRS Pharma LP, Patterson, New York, NY, USA), colloidal silicon dioxide (C-SiO_2_) (Aerosil^®^ 200 Pharma, Evonik Industries AG, Essen, Germany), and magnesium stearate (MgSt) (Mallinckrodt Inc., Raleigh, NC, USA) were used.

### 2.2. Sample Preparation

Uncoated tablets and powder blends were prepared in our laboratory, changing the formulation of the blends of five components (API, LM, MCC, C-SiO_2_, and MS). An unbalanced ratio of active pharmaceutical ingredients to excipients may compromise drug bioavailability and efficacy, resulting in adverse effects for patients. For that reason, this method was focused on monitoring the IBU concentration. Fourteen different concentrations were prepared in a range from 0% to 21% (*w/w*) IBU to develop a robust model that could detect failures in the process monitoring. The detection of placebo samples would indicate that the powder feeds were empty; in contrast, 21% (*w/w*) would represent that the drug content surpassed the concentration limits.

Conventional analytical methods are based on a calibration curve with samples of concentrations within ±30% of the target analyte. QCL is an emerging technology that needs to be validated to identify its limitations and capabilities. Therefore, we decided to cover a broad concentration range (0–21% API) to identify the linear dynamic range of this method. Having a better understanding of the QCL capabilities and being adopted as a routine analytical method, we expect future calibration samples to be capable of detecting a narrower range of concentrations within the API concentration target.

The blending process was performed using a digital mini vortex (Thermo-Fisher Scientific, Waltham, MA, USA) for 10 s at 3000 rpm. The powder blends were then pulverized in an agate mortar and mechanically mixed to ensure no agglomerate of particulate matter was present. The tablets were prepared using a manual laboratory press (Carver Standard Unheated Manual Press; Carver, Inc., Wabash, IN, USA). Compaction pressure was applied at 3000 psi (20.68 MPa).

### 2.3. QCL Instrument and Data Acquisition

A total of 280 spectra were obtained for both the samples of the tablets and powder blends (Table 1). For each concentration, 20 spectra were acquired at different locations within the sample surface using the back reflection mode of a LaserScan™ predispersive spectrometer (Block Engineering, Southborough, MA, USA) containing 3 tunable MIR lasers ranging from 990 to 1600 cm^−1^ as shown in Figure 1.

The individual laser tuning ranges were from 990 to 1111 cm^−1^, 1111 to 1178 cm^−1^, and 1178 to 1600 cm^−1^. The relative humidity and temperature remained at approximately 46% and 20 °C, respectively. A background spectrum of KBr was collected before any spectral data were acquired. Spectra were recorded using one scan. The time to acquire a complete scan was approximately 0.5 s for each laser, resulting in a total scan time of 1.5 s. The average power typically varied between 0.5 and 10 mW across the 600 cm^−1^ tuning range. Other laser parameters included a 100:1 transverse electromagnetic (TEM) polarization mode and a beam divergence of <2.5 and <5 mrad on the x- and y-axes. The fully contained spectrometer had a 3-inch-diameter ZnSe lens, which focused the MIR beam, collected the reflected light, and focused the light onto an internal thermoelectrically cooled mercury–cadmium–telluride detector. The spectroscopic system worked best at a head-to-target distance of 15 ± 3 cm. Each laser produced an elliptical image with 4 × 2 mm dimensions at the target’s focal plane (15 cm from the laser head) due to the difference in beam divergence in the axes.

### 2.4. MIR Multivariate Data Analysis

MIR vibrational data were obtained over the complete spectral range from 990 to 1600 cm^−1^. PLS regression models using the calibration data were developed and tested to quantify IBU in the tablets and powder blends. Several PLS models were generated based on the spectral range used using PLS Toolbox version 7.5.2 (Eigenvector Research Inc., Wenatchee, WA, USA) with the MATLAB R2011b version 7.13 platform (MathWorks, Natick, MA, USA). The performance of the calibration models was evaluated according to the following figures of merit: correlation coefficient (R^2^), root mean square error of calibration (RMSEC), root mean square error of prediction (RMSEP), root mean square error of cross-validation (RMSECV), and relative standard error of prediction (RSEP). Several PLS regression models were developed to quantify the IBU in the pharmaceutical formulations of the tablets and powder blends for each preprocessing treatment applied and each spectral analysis range. A few of these models were developed using the complete spectral range (990–1600 cm^−1^) and a combination of preprocessing treatments. The best models selected showed a similarity between the different parameters of the calibration and validation models, including RMSEC, RMSECV, and RMSEP. This maintains the robustness of the calibration and validation models for both the tablets and powder proposed. The RSEP is represented by the Equation (1):(1)$$RSEP\left(\%\right)=\sqrt{\frac{\sum_{j=1}^{n}{\left({\hat{y}}_{i,pred}-y_{i,ref}\right)}^{2}}{\sum_{j=1}^{n}{\left(y_{i,ref}\right)}^{2}}}\times 100\;$$

This equation indicates the prediction error of the model compared to the reference values (percentage). A simple inspection of the RSEP values can be used to determine which of the models is the best for each tablet and powder sample.

Several preprocessing treatments were applied to the spectroscopic data to reduce noise and highlight the spectral features related to chemical variation. These data pretreatments included a Savitzky–Golay (SG) first derivative (SG 1-D), a SG second derivative (SG 2-D), mean centering, multiplicative scattering corrections (MSC), standard normal variates (SNV), transformation to Log (1/R), and baseline corrections and their combinations.

### 2.5. Comparison of Spectral Noise FTIR and QCL

A comparison between diffuse reflectance FTIR (DR-FTIR) and diffuse reflectance QCL (DR-QCL) for C-SiO_2_, one of the most IR-absorbing excipients in the formulation, was performed to assess their performance in terms of spectral noise. Figure 2A shows the reflectance spectra of C-SiO_2_ in the spectral range from 1000 to 1600 cm^−1^. No significant differences were observed by visual inspection between DR-FTIR and DR-QCL for the acquired C-SiO_2_ spectrum. The sample does not exhibit any signal in the region from 1000 to 1300 cm^−1^; therefore, this spectral region was magnified (Figure 2B) for close visual noise inspection. It was found that the DR-QCL spectrum had a better spectral profile than the DR-FTIR spectra due to lower high-frequency noise. The calculations of the noise showed that the C-SiO_2_ spectra obtained by DR-FTIR had 30 times more noise than the DR-QCL C-SiO_2_ spectra; therefore, QCL spectroscopy provides less noisy spectra suitable for the characterization of APIs in formulations. A comparison between DR-FTIR and diffuse reflectance DR-QCL for IBU and other excipients is presented in Figure 3A–D, showing agreement. All signals and their corresponding positions were verified.

A comparison of the spectra for blends and tablets at concentrations 0 and 20% *w/w* was performed to identify the effects of the physical changes in the spectroscopic data shown in Figure 4A. The powders exhibited a weaker diffuse reflectance than the tablets. The compression forces exerted on the blends reduced the voids, making the tablet a compact solid with a smooth surface, reducing the diffuse transmission, and increasing the diffuse reflectance. This effect can be observed in the whole spectrum. The absorbance (−Log (R/R_0_)) for the powder blend showed a stronger signal than the tablet blend.

A weighted spectral subtraction (D) between the spectrum of the tablet and the spectrum of the powder blend was performed to find the differences in the tablet’s surface with respect to the powder formulation. The following equation was used to calculate the subtraction:(2)$$D\left[Y_{A}-Y_{B}\right]=Y_{A}-\left(\alpha +\beta \times Y_{B}\right)$$

*Y* is −Log (R/R_0_), and the subscripts A and B indicate the minuend and subtrahend spectra, respectively. The parameter α is the difference of the spectral offset between spectra *A* and *B*, and *β* is a compensation factor. The values of *α* and *β* can be calculated by classical least squares [31,32].

Figure 4B shows the spectral differences between tablets and powders. The difference observed is that the lactose signals are maintained in the 1000–1220 cm^−1^ region, highlighted in turquoise blue. The comparison is made with the pure lactose spectrum, indicating that the amount of this excipient on the tablet’s surface may have increased. A slope is also observed in the 1500–1600 cm^−1^ region, possibly apart from the broad band. This spectral feature can be attributed to MgST, which has a band with these characteristics [33,34]. Other signals did not match the pure component’s signals, which may be attributed to the interactions between them captured by this sensitive technique [30].

## 3. Results

### 3.1. Temporary Measures

The QCL spectra of a 10% IBU powder blend were measured 10 times at various areas in the blend and subsequently repeated after 15 and 60 days. The average spectra at 0, 15, and 60 days are shown in Figure 5A. Small changes in each spectrum were observed. To bring out these changes, a spectral subtraction was generated using Equation (1), where the average spectra at 15 and 60 days were subtracted from the average spectrum of day 0 (see Figure 5B). Various signals were observed, which could be due to the degradation of the components of the formulation.

Moreover, a strong signal of 1400 cm^−1^ at 60 days was observed, which should be attributed to the primary degradation product. Another possible explanation for these differences is the formulation of absorbed water vapor which generated strong interactions and significantly modified the infrared spectrum. This is evidenced by the discovery of lines when we subtracted the average spectrum from each of the spectra taken at different sites (see Figure 5A i); strong signs of water vapor were observed in the 1500–1600 cm^−1^ region at 15 and 60 days compared with day 0.

### 3.2. Partial Least Squares Models Results

Figure 6 shows the three best PLS regression models for each sample type (tablets and powder blends prepared from 0% to 20% IBU with excipients). The results are presented in terms of measured values (*w/w*%) versus predicted values (*w/w*%) for API. The tablet calibration and external validation sets are represented by red circles and black diamonds, respectively. The powder calibration set and external validation set are represented by red squares and black triangles, respectively. The dispersion of the calibration and validation data through the *y* = *x* line varies for the three different preprocessing methods selected, SNV, SG-1D + SNV, and SNV + MSC.

A model was developed with the raw spectra to obtain a reference model without pretreatment for further comparison. Considering the data dispersion, the best pretreatment for tablet calibration and validation models was SG-1D + SNV (Figure 6C). The best pretreatment for powder calibration and validation models was SNV and SNV + MSC, which were very similar (Figure 6E,F, respectively). Six models were selected in total, based on the statistical parameters summarized in Table 2.

An additional test was conducted to explore the reduction of the data acquisition time by developing PLS models and reducing the spectral range by using single QCL diodes. The best pretreatments previously discussed were applied using the complete spectral range/diode for the powder and tablet samples. Table 3 shows the parameters of RMSECV and RMSEP according to the type of sample, preprocessing, and diode used. The RMSECV and RMSEP values obtained for the QCL–PLS tablet and powder models using D_3_ are lower than D_1_ and D_2_. These results show that D_3_ contains a broader spectral region than the other two diodes, corresponding to more molecular information in the API’s vibrational signals.

However, the RMSEP value of the models obtained using the spectral region of D_3_ for the tablets and powder blends is slightly worse than the RMSEP value obtained using all spectral regions. Therefore, the remaining spectral region of D_3_ (990–1178 cm^−1^) contributes to improving the multivariate model. The characteristic bands of the API are observed in this region.

### 3.3. Effect of Depth of Focus on Predictability

The distance from the target to the detector is a significant factor when performing remote detection measurements in an online analysis. The QCL system was focused on a 15 cm depth, representing a 0 cm depth of focus as shown in Figure 7.

The distance between the QCL spot and the target was adjusted on the z-axis (depth) to defocus the system and then measure the effects of those changes in the error of prediction (% error) of the sample concentration. A sample with 10 (*w/w*%) API was prepared, and the depth of the z-coordinate was changed from 6 to 22 cm. The best depth of focus conditions for a correct prediction of this sample were ±3 (corresponding to 15 ± 3 cm), with a ± 10% prediction error. For the depth of focus values above 3 or below −3 cm, the prediction of the API concentration in the sample was erroneous, and the deviation from the reference value of 10 *w/w*% API was significant.

## 4. Conclusions

Quantum cascade laser technology can significantly impact the pharmaceutical manufacturing of oral solid dosages due to QCL’s high signal-to-noise spectra with a short integration time, enabling the monitoring of the fast dynamics process with high confidence. Sudden changes in the process will be detectable with this system, considering that the system requires only 1.5 s to acquire a spectrum.

The MIR region provides a unique absorption spectrum for different components; it also enables the detection of polymorphic transformation. The analytical method based on QCL laser spectroscopy successfully quantified the API at concentrations (0–21% *w/w*) in tablets and powders. Therefore, the system could detect the API in complex matrices of several excipients with high specificity. The PLS models developed to quantify the API from tablets and powders required only the first derivative and SNV as preprocessing steps, yielding excellent R^2^ and RMSEP values.

The MVA results of the QCL–PLS models obtained by individual diodes showed that the best results were obtained using D_3_ (1178–1600 cm^−1^). This diode covers a more extensive spectral range, where the vibrational signatures of the API are present. Therefore, a QCL using only this diode would reduce the acquisition time of the spectra.

Conventional FTIR spectrometers are well-established instruments, but QCL has a higher spectral power density. The spectral analysis of C-SiO_2_ in the spectral range from 1000 to 1600 cm^−1^ was performed using two techniques, DR-FTIR and MIR laser DR-QCL, showing that QCL has a better signal-to-noise ratio, which suggests that it is helpful for online sensing.

The best depth of focus conditions for a correct API prediction were 15 ± 3 cm, with a ±10% prediction error; therefore, implementing this method for online sensing should involve careful adjustment of the experimental setup to focus the laser. Thus, implementing this approach in a continuous manufacturing line must involve consideration of the distance of the sample from the laser during the calibration measurements to make a robust system method that accounts for these variations, particularly for the powder blends.

The QCL system provides a highly collimated laser spot, resulting in a high spatial resolution suitable for surface mapping to evaluate the distribution of the pharmaceutical components in the tablet. Researchers have demonstrated that photothermal imaging of pharmaceutical tablets using quantum cascade lasers provides information on the distribution of the API and excipients [35]. The detection limits of this method were reported to be 1% *w/w* in previous studies [30]; therefore, implementing this method for cleaning validation of the equipment in pharmaceutical sites would reduce downtime.

Finally, the integrity of the pharmaceutical formulations was evaluated by temporary measurements that confirmed the degradation or moisture absorption in the formulations. The versatility of QCL to analyze liquid, solid, and gas samples makes it suitable for pharmaceutical operations as it is portable, powerful, and provides high specificity.

## Figures and Tables

**Figure 1 pharmaceutics-13-00985-f001:**
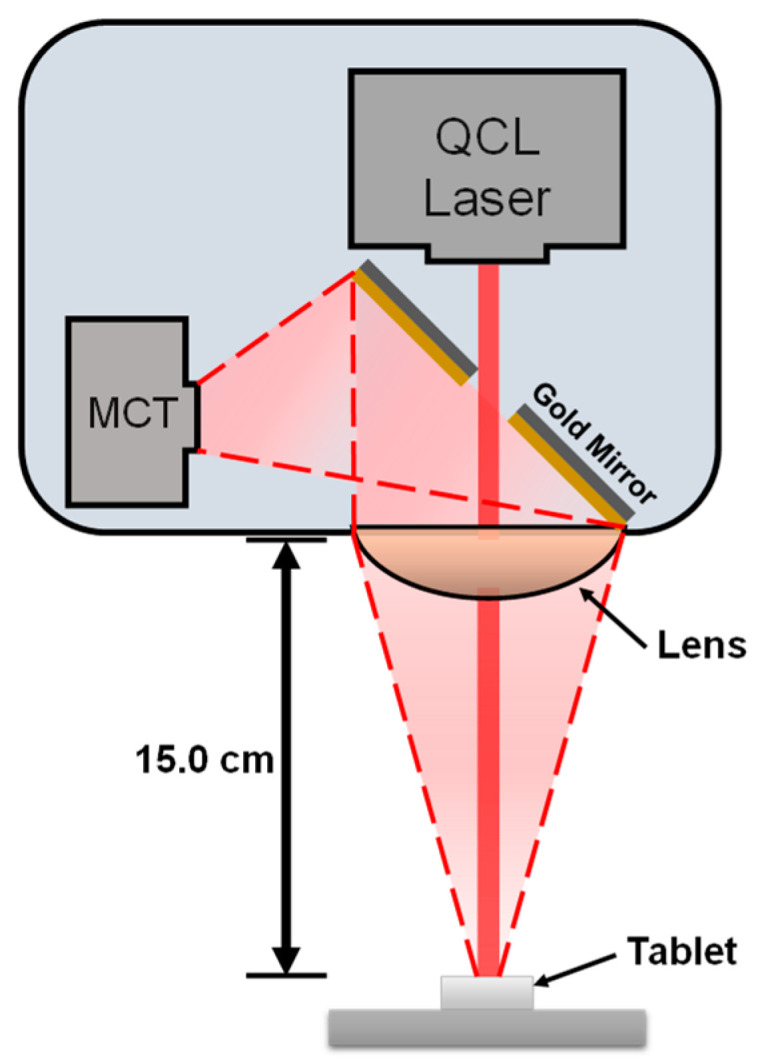
Schematic of diffuse reflectance quantum cascade laser experimental setup.

**Figure 2 pharmaceutics-13-00985-f002:**
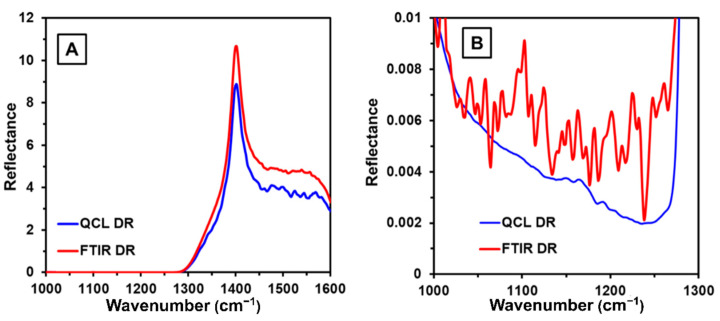
Comparing the spectroscopic instruments and modes (DR, diffuse reflectance; TR, transmission) used for the characterization of C-SiO_2_: FTIR and QCL. (**A**) Spectral range from 1000 to 1600 cm^−1^; (**B**) zoom of the spectral range from 1000 to 1300 cm^−1^.

**Figure 3 pharmaceutics-13-00985-f003:**
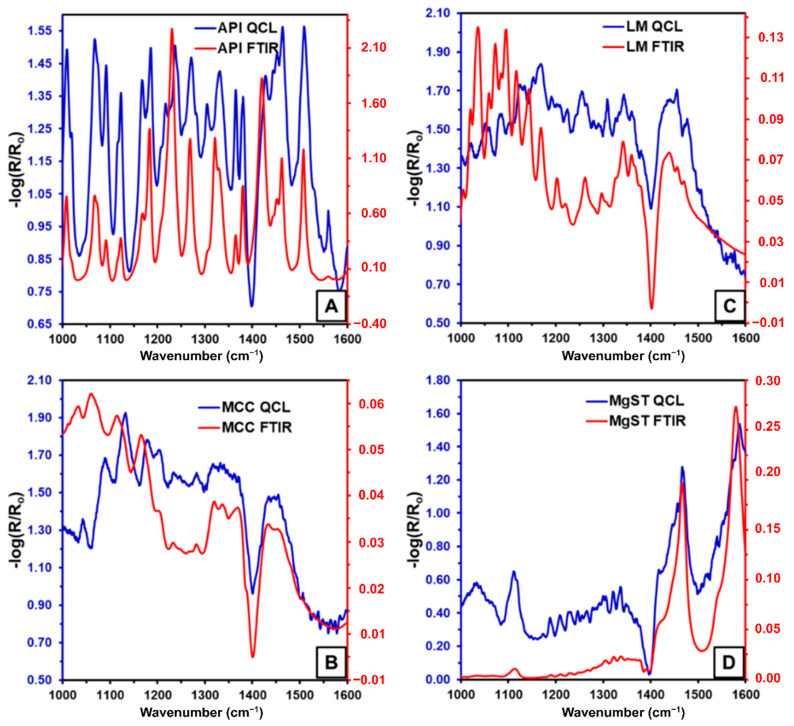
Comparison of the QCL FTIR diffuse reflectance spectra of pure components (**A**) IBU, (**B**) ML, (**C**) MCC, and (**D**) MgST.

**Figure 4 pharmaceutics-13-00985-f004:**
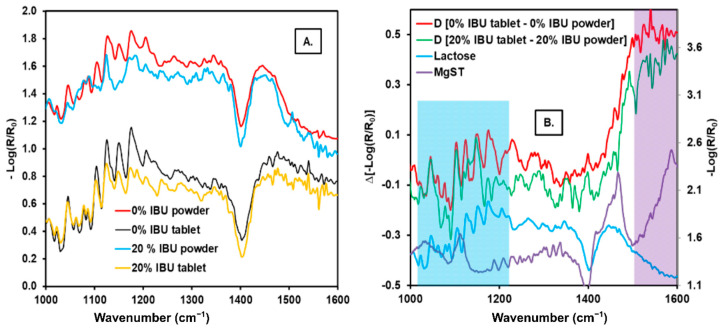
(**A**) Comparison of the QCL reflectance spectra of the tablets and powders with 0% and 20% API and (**B**) weighted spectral subtraction between the tablet spectrum and that of the powder with 0% and 20% API.

**Figure 5 pharmaceutics-13-00985-f005:**
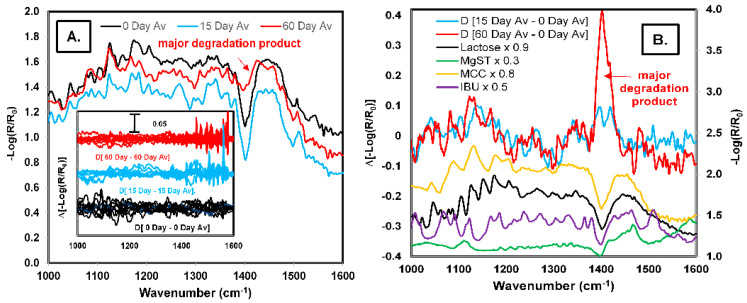
(**A**) Comparison between the average QCL reflectance spectra of the powder with 10% API to different times, figure inserted i (spectral subtraction between the 10 spectra of powder with 10% API in different sites and the average spectrum for each time), figure inserted (spectral subtraction between the 10 spectra of the powder with 10% API in the same site and the average spectrum). (**B**) Comparison between spectral subtraction of the spectra at other times and time zero and the components of the formulation.

**Figure 6 pharmaceutics-13-00985-f006:**
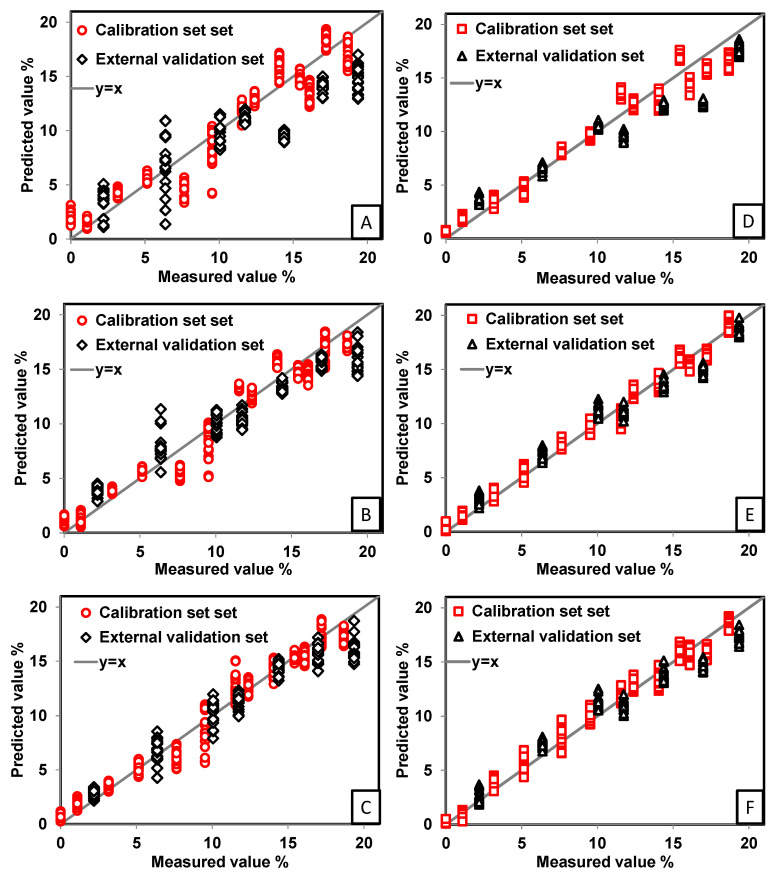
PLS regressions obtained for the measured API values in % *w/w* versus the predicted IBU values in % *w/w* for the calibration set and the external validation set. (**A**) API prediction in tablets with no preprocessing; (**B**) API prediction in tablets with SNV preprocessing; (**C**) API prediction in tablets with SG-1D + SNV preprocessing; (**D**) API prediction in powder blends with no preprocessing; (**E**) API prediction in powder blends with SNV preprocessing; and (**F**) API prediction in powders blend with SNV + MSC preprocessing.

**Figure 7 pharmaceutics-13-00985-f007:**
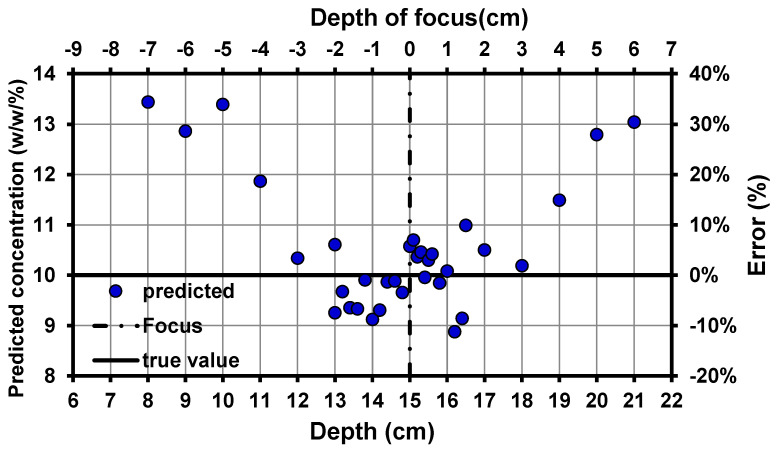
Variation in the percentage of error (%) and predicted concentration (%, *w/w*) as a function of the depth of focus for the QCL beam in predicting a sample with 10% *w/w* API.

**Table 1 pharmaceutics-13-00985-t001:** Composition of the sample sets used to develop the QCL method used to detect the API in powder and tablet samples.

Type of Sample	Tablets/Powder Blends
Total number of samples	21
Number of QCL spectra per sample	20
Calibration set	14
External validation set	7
API concentration level for the calibration set	0.00%, 1.10%, 3.17%, 5.15%, 7.65%, 9.53%, 11.57%, 12.39%, 14.06%, 15.44%, 16.10%, 17.22%, 18.68%, and 21.67%
API concentration level for the external validation set	2.20%, 6.37%, 10.06%, 11.74%, 14.38%, 16.98%, and 19.36%

**Table 2 pharmaceutics-13-00985-t002:** MVA results obtained for tablets and powder blends using the PLS models of QCL reflectance data.

Calibration Test					
Sample	Preprocess	RMSEC (%)	RMSECV (%)	R^2^CV	BiasCV
Tablets	None	1.68	1.69	0.933	−0.02
	SNV	1.33	1.38	0.955	−0.0009
	SG-1D + SNV	1.13	1.16	0.968	−0.004
Powder	None	1.33	1.34	0.958	−0.06
Blend	SNV	0.75	0.77	0.986	−0.001
	SG-1D + SNV	0.767	0.787	0.985	0.0002
**Test Set**					
**Sample**	**Preprocess**	**R^2^Pred**	**RMSEP (%)**	**RSEP (%)**	**LVs**
Tablets	None	0.825	3.11	2.93	2
	SNV	0.942	1.95	1.16	3
	SG-1D + SNV	0.963	1.38	0.58	4
Powder	None	0.928	2.15	1.42	2
Blend	SNV	0.973	1.16	0.41	3
	SG-1D + SNV	0.972	1.18	0.43	3

**Table 3 pharmaceutics-13-00985-t003:** Multivariate analysis results obtained for QCL–PLS models by using individual diodes.

Sample	Spectral Preprocessing	Diodes	Spectral Region (cm^−1^)	RMSECV (%)	RMSEP (%)
Tablets	SG-1D + SNV	D_1_	990–1111	2.05	2.49
		D_2_	1111–1178	1.82	2.28
		D_3_	1178–1600	1.33	1.8
Powder	SNV	D_1_	990–1111	1.29	2.51
		D_2_	1111–1178	1.19	2.56
		D_3_	1178–1600	0.92	1.33

## Data Availability

Not applicable.

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
