# Peer review of "API Content and Blend Uniformity Using Quantum Cascade Laser Spectroscopy Coupled with Multivariate Analysis"

_pharmaceutics, 2021, doi:10.3390/pharmaceutics13070985_

Round 1

Reviewer 1 Report

Comments on Pharmaceutics-1212337

The authors presented a novel method to quantify ibuprofen using a MIR quantum cascade laser spectroscopy. Overall this manuscript is of good quality, and the protocol developed by the authors will have impacts on current pharmaceutical process. However, some concerns need to be address before being published.

The major one is the authors put too many method details in the results part, making the reviewer and the readers difficult to follow what exactly experiments have been done. For example, Line 149-204, first paragraph of 3.3, line 260-267, these parts should be removed back to Materials and methods.

Another concern is the authors only presented the results, but not provided sufficient discussion, such as the advantages, limitation, other literature reports, and current process about this technique. Also, the authors are suggested to discuss more about the potential impact and the importance of applying this technique in pharmaceutical fields.

Minor concerns:

Figure 2 (B), please considering removing FTIR-TR since this panel was meant to compare QCL DR and FTIR DR.

Please provide rationale of choosing ibuprofen and setting the concentrations from 0-21%. Also, this study only tested IBU one compound, what about others? The authors may need to discuss this as well.  

Author Response

The document was modified following the suggestions of the reviewer

See attached file

Reviewer 2 Report

The aim of the paper is to propose a reproducible methodology for the determination of ibuprofen in formulations of powder blends and tablets using mid-infrared quantum cascade laser spectroscopy-based protocol with the additional use of partial least squares models to estimate the correlation of vibrational signals and active pharmaceutical ingredients concentrations of powders and tablets. The manuscript is organized and well written. The conclusions are consistent with the results obtained and with the data already existing in literature. The findings are also well discussed. In my opinion, the submission requires minor edition and improvement at some points, main of which are listed below.

Minor issues:

L75-77, 123, 138, 308: please enter missing spaces between words or between words and the quoted literature.

L241-243: Where in figure 6 black diamonds and black triangles are marked?

L251: please correct the font size in the sentence.

Round 2

Reviewer 1 Report

Revised manuscript can be accepted with present form. 

Author Response

The English revision was done and small changes were made. I adjust past corrections
